# WWC Proteins: Important Regulators of Hippo Signaling in Cancer

**DOI:** 10.3390/cancers13020306

**Published:** 2021-01-15

**Authors:** Verena Höffken, Anke Hermann, Hermann Pavenstädt, Joachim Kremerskothen

**Affiliations:** Internal Medicine D, Department of Nephrology, Hypertension and Rheumatology, University Hospital Münster, 48147 Münster, Germany; verena.hoeffken@ukmuenster.de (V.H.); hermann@neuway-pharma.com (A.H.); hermann.pavenstaedt@ukmuenster.de (H.P.)

**Keywords:** WWC, Hippo pathway, cancer, EMT, YAP, LATS1/2, AMOT, aPKC

## Abstract

**Simple Summary:**

The conserved Hippo pathway regulates cell proliferation and apoptosis via a complex interplay of transcriptional activities, post-translational protein modifications, specific protein–protein interactions and cellular transport processes. Deregulating this highly balanced system can lead to hyperproliferation, organ overgrowth and cancer. Although WWC proteins are known as components of the Hippo signaling pathway, their association with tumorigenesis is often neglected. This review aims to summarize the current knowledge on WWC proteins and their contribution to Hippo signaling in the context of cancer.

**Abstract:**

The Hippo signaling pathway is known to regulate cell differentiation, proliferation and apoptosis. Whereas activation of the Hippo signaling pathway leads to phosphorylation and cytoplasmic retention of the transcriptional coactivator YAP, decreased Hippo signaling results in nuclear import of YAP and subsequent transcription of pro-proliferative genes. Hence, a dynamic and precise regulation of the Hippo signaling pathway is crucial for organ size control and the prevention of tumor formation. The transcriptional activity of YAP is controlled by a growing number of upstream regulators including the family of WWC proteins. WWC1, WWC2 and WWC3 represent cytosolic scaffolding proteins involved in intracellular transport processes and different signal transduction pathways. Earlier in vitro experiments demonstrated that WWC proteins positively regulate the Hippo pathway via the activation of large tumor suppressor kinases 1/2 (LATS1/2) kinases and the subsequent cytoplasmic accumulation of phosphorylated YAP. Later, reduced WWC expression and subsequent high YAP activity were shown to correlate with the progression of human cancer in different organs. Although the function of WWC proteins as upstream regulators of Hippo signaling was confirmed in various studies, their important role as tumor modulators is often overlooked. This review has been designed to provide an update on the published data linking WWC1, WWC2 and WWC3 to cancer, with a focus on Hippo pathway-dependent mechanisms.

## 1. Introduction

### 1.1. Hippo Signaling and Cancer

The Hippo signaling pathway and its impact on cell proliferation, apoptosis and organ growth have been examined with increasing interest within the last two decades (reviewed in [1,2,3]). The first components and regulatory mechanisms of the Hippo signaling pathway were characterized in invertebrate model systems. In *Drosophila melanogaster*, the loss of the kinase *Warts* (*Wts*) was shown to induce hypertrophy of epithelial cells, indicating a role of *Wts* in organ size control [4]. In the same year, the group of Wu et al. characterized the Ste-20 family Ser/Thr kinase *Hippo* (*Hpo*) and described a tumor-suppression pathway in which a kinase cascade consisting of *Hpo*, *Salvador* (*Sav*, an adaptor protein for *Hpo*) and *Wts* regulates cell proliferation and apoptosis by upregulating the transcription of the cell cycle regulators *Diap1* and *CycE10* [5]. Furthermore, the transcriptional coactivator *Yorkie (Yki)* was identified as the missing link between *Wts* activity and transcriptional regulation [6].

As the Hippo pathway is highly conserved from invertebrates to mammals, most pathway components in invertebrates have mammalian orthologs [7]. Upon activation via upstream signals, the mammalian Ste-20-like protein kinases 1/2 (MST1/2, *Hpo* in *Drosophila*) phosphorylate the large tumor suppressor kinases 1/2 (LATS1/2, *Wts* in *Drosophila*). The activity of MST1/2 is further enhanced by their activator WW45 (*Sav* in *Drosophila*). Activated LATS1/2, in combination with their adaptor proteins MOB kinase activator 1 A and B (MOB1A/B, *Marts* in *Drosophila*), phosphorylate the transcriptional coactivator Yes-associated protein (YAP; *Yki* in *Drosophila*) and its paralog, the transcriptional coactivator with PSD-95, Discs-large, ZO-1 (PDZ)-binding motif (TAZ, also known as WW domain-containing transcription regulator 1 (WWTR1)). As long as YAP and TAZ are phosphorylated, they are either trapped in the cytoplasm via an interaction with members of the 14-3-3 protein family or are degraded via the proteasome upon ubiquitination. In contrast, unphosphorylated YAP and TAZ are able to enter the nucleus and to bind to a group of transcription factors, including TEA domain family member 1 (TEAD1). This interaction, in turn, induces the expression of pro-proliferative genes (reviewed in [1,2,3]). Thus, it can be concluded that the activity of the MST1/2 and LATS1/2 kinases as well as the phosphorylation status of YAP and TAZ constitute the main regulatory switches controlling Hippo signaling.

As cytosolic localization and post-translational modification of YAP and TAZ are crucial for Hippo signaling, cellular mechanisms regulating these processes have been intensively studied within the last years. Furthermore, the group of upstream regulators for Hippo signaling is continuously growing, consisting of cytoskeleton proteins, membrane receptors, enzymes and scaffolding proteins (reviewed in [1,2,3,8]).

### 1.2. WWC Proteins: Upstream Regulators of the Hippo Pathway

Homeostasis in Hippo signaling includes regulated protein–protein interactions and the formation of large membrane-associated multiprotein complexes. Members of the WWC and C2 domain-containing (WWC) protein family were described as scaffolding proteins in different signaling pathways (reviewed in [9,10]). By recruiting and anchoring protein complexes to specific cellular subdomains, WWC proteins were shown to have an important impact on cell proliferation, cell polarity or vesicle transport [11,12,13]. WWC1, also named KIBRA because of its relatively high expression in kidney and brain, was originally identified as an interaction partner of the postsynaptic protein Dendrin in a yeast two-hybrid screen using a human brain library [14]. The human *WWC1* gene encodes a cytosolic protein of about 125 kDa that displays a perinuclear distribution [14]. Later, the highly similar proteins WWC2 and WWC3 were identified as additional members of the human WWC protein family [15].

Studies using quantitative reverse transcriptase polymerase chain reaction (qRT-PCR) demonstrated a similar tissue expression pattern of the different WWC mRNAs, with high levels in lung and moderate expression in brain, kidney and liver [15]. Interestingly, WWC2 is prominently expressed in the testis, whereas WWC3 displays high expression in ovaries [15].

Like the Hippo signaling itself, WWC proteins are also highly evolutionarily conserved among species. Phylogenetic studies revealed that the WWC protein family arose from a single WWC1-like sequence (*kibra*) already present in invertebrates such as *Drosophila* [15]. In fish, a common progenitor of the later *WWC1/WWC2* genes and a *WWC3*-like gene have been found. Finally, the mammalian genome contains genes encoding all three members of the protein family, WWC1, WWC2 and WWC3. An exception is the house mouse (*Mus musculus*), in which the *Wwc3* gene was deleted from the genome during evolution [15].

Because of the evolutionary ancestry of the WWC family members, it is not surprising that they share common features and characteristics at the structural level. All mammalian WWC proteins are composed of two amino-terminal WW domains and an integral C2-like domain (Figure 1). The WW domains, named after the presence of two characteristic tryptophan (WW) residues, mediate binding to proline-rich motifs (PPxY or LPxY) in interaction partners (reviewed in [9,10]). The C2-like domain of the WWC proteins binds to phospholipids and mediates their localization to specific membrane structures and cellular vesicles [16]. Moreover, all WWC proteins characteristically contain coiled-coil domains (CC), an atypical protein kinase C (aPKC)-binding motif (aPKC BM) and a PDZ-binding motif (PDZ BM) (Figure 1) (reviewed in [9,10]). Unique to WWC1 is a poly glutamic acid (polyE) stretch with unknown function. A proline-rich region is localized at the amino-terminus of WWC3, which might mediate an interaction with specific binding partners (Figure 1). As WWC proteins form homo- and heterodimeric head-to-tail complexes, they are supposed to provide a platform for large multiprotein complexes [15].

Beside specific protein–protein interactions, post-translational phosphorylation was demonstrated to control WWC1 activity in cell migration and proliferation. Human WWC1 can be phosphorylated at serine 539 (S539) by the kinases Aurora A/B, at S548 by the extracellular signal-regulated kinase (ERK), at S975/978 by atypical protein kinase C (aPKC) and at threonine 929 (T929) and S947 by the 90-kDa ribosomal S6 kinase (RSK) (reviewed in [10]). Like the other structural domains, those sequence motifs containing these phosphosites are conserved and can also be found in WWC2 and WWC3 (Figure 1).

The common expression pattern and the structural similarities within the WWC protein family indicate a synergistic or redundant function of the individual family member in individual cells or tissue. Indeed, Hermann et al. could show that a cell-specific loss of Wwc1 in mouse liver hepatocytes is compensated by Wwc2, and vice versa [17]. Furthermore, induced overexpression of human WWC1, WWC2 or WWC3 in *Drosophila* leads to comparable effects on organ growth, indicating an overlapping function of the different WWC proteins [15].

First data about the potential role of the WWC proteins in Hippo signaling were published independently from different groups in 2010 [18,19,20]. In these studies, *Kibra* was shown to interact with *Merlin* (*Mer*) and *Expanded* (*Ex*), known upstream regulators of the Hippo pathway in *Drosophila*. Inhibition of *Kibra* expression by RNA interference led to enhanced activity of *Yki*, increased cell proliferation and organ overgrowth.

In mammalian cells, WWC proteins were shown to interact with LATS1/2 kinases [21]. Binding to WWC1 stimulates the phosphorylation of LATS1/2, increasing their enzymatic activity. Subsequently, LATS1/2-mediated phosphorylation of YAP is enhanced, preventing its nuclear entry and hindering transcription of pro-proliferative genes [21]. In line with this, Wennmann et al. could show that WWC overexpression leads to a Hippo-pathway-dependent decrease in cell proliferation [15].

## 2. WWC Proteins and Cancer

Since WWC1 was linked to Hippo pathway-dependent cell proliferation and organ size control 10 years ago, interest grew on a putative role of WWC proteins in tumor formation. Initially, studies addressed the impact of WWC1 in human cancer, trying to uncover the involved pathways, specific protein interactions, regulation mechanisms and possible target points for new anti-cancer therapies (reviewed in [9,10]). Until today, the association of WWC1 and cancer formation has been confirmed in more than 80 publications (Figure 2).

Considering the structural similarities and putative redundant functions of all three WWC protein family members, WWC2 and WWC3 are supposed to play a WWC1-comparable role in human cancer. However, publications presenting data about the correlation of WWC2/WWC3 function and tumorigenesis are limited in number (Figure 2).

Obviously, WWC-focused studies represent only a very small percentage of all approaches aiming to elucidate the role of Hippo signaling itself in cancer (more than 2300 PubMed-listed publications from 2010 to 2020). However, there is growing evidence for the crucial role of the WWC proteins in Hippo signaling-dependent hyperproliferation and tumorigenesis. The published data also indicate that WWC proteins affect the crosstalk between different signal transduction pathways involved in tumorigenesis. In the following chapters, we will summarize the present findings about the association of WWC function and cancer with a focus on Hippo pathway-related mechanisms.

### 2.1. WWC1

The impact of WWC1 function on cellular hyperproliferation and tumorigenesis appears to be controlled at the genomic, transcriptional and translational levels. However, the data about the role of WWC1 in cancer are contradictory. In several investigations, WWC1 was shown to act as a tumor suppressor, and a disturbed WWC1 expression was often correlated to decreased Hippo pathway signaling and YAP-mediated increase in cell proliferation. In contrast, other studies revealed a correlation between an increase in WWC1 expression in distinct human cancer types, pointing to a more pro-proliferative role of the protein.

Most of the publications in this field link WWC1 function to human breast cancer (BC). Earlier findings displayed that in normal breast tissues, WWC1 plays a role in the development of mammary glands. Hilton et al. could demonstrate that WWC1 expression varies during the gland development and is controlled by progestin and prolactin in epithelial breast cells [22]. A lack of WWC1 expression leads to a disturbed mammary gland development frequently linked to BC. WWC1 function in breast tissue likely depends on the activity of the discoidin domain receptor 1 (DDR1). DDR1 forms a trimeric complex with WWC1 and aPKC, suppressing mitogen-activated protein kinase (MAPK)/extracellular-regulated kinase (ERK) signaling and breast cell proliferation. Stimulation of DDR1 phosphorylation by collagen leads to dissociation of the trimeric complex and enhanced MAPK/ERK signaling, promoting tumor formation [22].

A study by Mussell et al. describes WWC1 as a suppressor of epidermal growth factor (EGF) receptor signaling in BC. Disturbed WWC1 expression forces epidermal-to- mesenchymal transition (EMT) and nuclear YAP activity to induce amphiregulin (AREG) expression. Subsequently, AREG activates the EGF receptor, which leads to enhanced proliferation and migration of BC cells, ending in poor survival and recurrence of BC for the patient [23].

BC subtypes show a specific cell surface expression of hormone receptors for estrogen (ER), progesterone (PR) and human epidermal growth factor receptor 2 (HER2) (reviewed in [24]). This expression pattern in combination with additional markers, e.g., for EMT or stem cell proliferation, is often used for BC classification [25]. Different localization and the expression level of WWC1 could be associated with specific BC subtypes. While a nuclear expression of WWC1 is associated with good prognostic features (low recurrence, lymph node stage and positive ER/PR expression), a cytoplasmic expression indicates poor prognostic features with HER2 overexpression and high-grade tumors [25]. An overall low WWC1 expression correlates with negative ER/PR expression and predicts recurrence-free survival disadvantages for BC patients after an endocrine therapy [25]. Low WWC1 expression level is also significantly associated with ER-negative/PR-negative, high-grade tumors and a higher risk of relapse and death [26]. Mechanistically, data from Rayala et al. revealed that WWC1, in complex with dynein light chain 1 (DLC1), induces ER activity, thus increasing the estrogen-sensitive expression of ER target genes, which, in turn, promote the proliferation of BC cells [27].

Triple-negative breast cancer (TNBC) represents a highly invasive and aggressive BC subtype that is defined by the lack of ER, PR and HER2 expression (reviewed in [28]). Chromosome 5q instability was identified as a tremendous factor in TNBC, and in 70% of all TNBC cases, the 5q arm harboring the WWC1 gene locus is deleted [28]. Studies from Knight and colleagues correlated the loss of the WWC1 locus with the promotion of tumor progression and metastasis in TNBC [28]. Indeed, inhibiting WWC1 expression results in enhanced YAP and TAZ activity and increased proliferation and invasion of TNBC cells, augmenting their tumorigenic and metastatic potential [29].

Since WWC1 is highly expressed in the kidney, the protein was thought to play a potential role in renal cancer subtypes. Accordingly, in patients suffering from clear-cell renal cell carcinoma (ccRCC), WWC1 expression is often epigenetically silenced [30]. Methylation of the WWC1 promotor at two CpG islands is enhanced in ccRCC patients, resulting in significantly reduced WWC1 mRNA and protein levels [30]. A putative linkage between WWC1 expression, Hippo pathway activity and ccRCC has not been analyzed, so far.

Epigenetic regulation of WWC1 expression was also observed in gastric cancer (GC) [31]. The level of WWC1 promotor methylation in GC patients showed correlations with patients’ age (≥60 years), increased GC risk and environmental factors [31]. Furthermore, high expression of WWC1 in GC samples expressing low aPKC correlates with higher lymphatic and venous invasion, patients’ age (≥70 years) as well as reduced 5-year disease- and relapse-free survival for the patient [30]. Due to enhanced WWC1 expression, a loss of aPKC activity was found to result in a deficiency of intercellular junctions and increased invasiveness of GC cells [32].

A recent case–control study discovered WWC1 expression to be increased in bladder cancer (BLC) patients [33]. Patients carrying a specific single-nucleotide polymorphism (SNP) within the WWC1 gene had a decreased risk for BLC [33]. However, it is unclear how this SNP affects WWC1 activity in Hippo signaling in BLC.

In colorectal cancer (CRC), enhanced cell proliferation and migration were found to be associated with a negative regulation of WWC1 expression caused by the transcription factor TCF19. Due to TCF19 inhibition, the tumor-suppressive function of WWC1 is prevented, which, in turn, increases metastasis and malignant progression, depleting the survival rates for CRC patients [34].

Interestingly, the presence of a fusion gene that arose from a partial chromosomal deletion of the instable chromosome region 5q and contains parts of the WWC1 and the FAM174A genes leads to an increased invasiveness and oncogenic capacity in early onset CRC [35]. Due to the gene fusion, WWC1 function was impaired, Hippo pathway activity was low and nuclear YAP level increased. As a result, metastasis, tumor sphere propagation and oncogenic capacity were significantly increased [35].

In osteosarcoma, the transcription factor SRY-Box Transcription Factor 2 (Sox2) was shown to counteract the Hippo pathway and to restrain its two activating components, WWC1 and neurofibromin 2 (NF2, *merlin* in *Drosophila*) [36]. Due to this inhibitory effect on WWC1 and NF2 transcription, enhanced YAP activity induces the expression of target genes that are important to maintain the cancer stem cell characteristics and differentiation in osteosarcoma.

The role of WWC1 in hematopoietic cancer is poorly elucidated. Epigenetic regulation of WWC1 expression was shown to correlate with progression of acute lymphocytic leukemia (ALL) [37]. In the majority of B-cell ALL, the *WWC1* gene is methylated, resulting in low expression [37]. Epigenetic control of WWC1 expression also occurred in chronic lymphocytic leukemia (CLL), which is defined as a lymphoma with B cell accumulation in blood, bone marrow and lymph nodes [38]. In about one third of samples from CLL patients, the *WWC1* gene was methylated and a reduced WWC1 expression was detectable. Low WWC1 levels were associated with negative biological parameters including poor outcome [38]. As YAP and TAZ are dispensable for physiological and malignant hematopoiesis [39], WWC1 function in hematopoietic cancer is likely Hippo pathway-independent.

Decreased WWC1 expression was linked to lung adeno carcinoma (LUAD), which is a subtype of non-small-cell lung cancer (NSCLC). An elevated level of microRNA-21 (miR-21) was found to inhibit WWC1 synthesis and, subsequently, Hippo pathway activity. Thus, an miR-21-induced decrease in WWC1 protein promoted cell proliferation, migration and metastasis in LUAD while apoptosis was impeded, resulting in tumor progression [40].

In contrast, a significantly increased expression of WWC1 was found in prostate cancer (PC) tumors and corresponding cell lines [41]. High WWC1 levels were linked to elevated proliferation, migration and invasion of PC cells. A second study further confirmed the role of WWC1 as a promotor of metastasis in PC [42]. The authors demonstrated that WWC1 is able to induce the formation of a trimeric complex with the polarity regulators Par3 and aPKC. As a result, Hippo pathway activity is decreased, leading to an enhanced YAP-dependent expression of pro-metastatic genes for PC [42]. It is worth mentioning that a ubiquitous Wwc1 knockout in mice causes a mild neurological phenotype but does not lead to organ overgrowth or tumor formation [17]. An explanation for this might be a compensatory effect of WWC2 in organs lacking WWC1 expression.

### 2.2. WWC2

Since its first description in 2015, only a very limited number of studies have been published that investigated the expression and function of the WWC2 protein in tumorigenesis (Figure 2). However, due to its high structural similarity to WWC1 and a comparable expression pattern, WWC2 is supposed to fulfil a regulatory role in the Hippo signaling pathway and human cancer, too.

In mouse hepatic cells, a complete loss of WWC expression induces Hippo pathway-dependent hepatocyte transdifferentiation, activation of inflammation, fibrosis and, finally, hepatocellular carcinoma (HCC) [17]. Interestingly, a Wwc1 knockout alone does not lead to hepatic tumor formation but requires the simultaneous inactivation of Wwc2 expression [17]. In accordance with this, expression of WWC2 in human HCC was found to correlate negatively with nuclear YAP levels but positively with phosphorylated YAP in the cytoplasm [43]. WWC2 was also shown to suppress EMT, the invasion of tumor cancer cells and metastasis of HCC. Evaluation of patients’ data demonstrated that high WWC2 expression correlated with better survival, whereas low expression of WWC2 was associated with advanced clinicopathological features of HCC and poor prognosis [43].

Wang et al. described a miRNA-dependent regulation of WWC2 protein level in pancreas cancer (PAC) [44]. They analyzed the role of miR-10a in EMT and stemness maintenance of pancreatic cancer stem cells (PCSCs) and identified a set of differentially expressed miR-10a target genes, including *WWC2*. Furthermore, in PAC tissues, miR-10a was highly expressed, whereas WWC2 level was low. Inhibition of miR-10a transcription reduced the EMT and stemness maintenance of PCSCs by upregulating WWC2 and subsequent Hippo signaling pathway activation [44].

The miRNA-dependent regulation of WWC2 expression could also be observed in LUAD. Whereas expression of miR-21-5p was upregulated in LUAD tissues, WWC2 protein level was low and was associated with poor prognosis [45]. Thus, miRNA-21-5p promoted cell proliferation, migration and invasion in LUAD via downregulation of WWC2.

A recent study on colorectal cancer (CRC) investigated the effect of the long non-coding RNA LINC00460 on WWC2 gene transcription and Hippo signaling [46]. By binding to the WWC2 promotor region and recruiting the transcription factor ETS-related gene (ERG), LINC00460 prevents *WWC2* mRNA synthesis. Consequently, low *WWC2* transcription inhibits Hippo signaling and forces YAP-dependent transcription of the downstream target genes Snail, TWIST1 and N-cadherin, promoting enhanced CRC cell migration, invasion, EMT and CRC progression [46].

Frassanito et al. demonstrated that multiple myeloma (MM), a disease associated with tumor plasma cells in bone marrow, utilizes an exosome-dependent regulation of WWC2 level for Hippo pathway control and subsequent manipulation of the tumor microenvironment [47]. They could show that exosomes containing WWC2 protein are released from MM cells and fuse with surrounding fibroblasts. Increased WWC2 level in targeted fibroblasts leads to the accumulation of phosphorylated YAP in the cytoplasm, where it activates miRNA synthesis. Thus, MM cells modify the miRNA profile of fibroblasts by increasing miR-27b and miR-214 expression, which, in turn, supports paracrine growth of plasma cells and tumor progression [47].

### 2.3. WWC3

First papers investigating the function of WWC3 in malignant diseases were published in 2017 (Figure 2). Analysis of lung cancer cell lines and NSCLC specimens demonstrated that WWC3 expression was significantly downregulated and correlated with weak cell differentiation, positive lymph node metastasis, advanced tumor–node–metastasis stage as well as poor prognosis and shortened survival in lung cancer patients [48]. In NSCLC, reduced WWC3 expression leads to low Hippo pathway activity and increased nuclear YAP levels. Enhanced transcriptional activity of YAP causes an increase in colony formation, proliferation and invasion of lung cancer cells. In line with these findings, induced WWC3 overexpression in a tumor mouse model led to decreased cell proliferation and reduced lung weight [48].

Han et al. showed that low WWC3 expression in lung cancer cell lines promotes EMT. The authors demonstrated that WWC3 expression correlated positively with epithelial markers (zona occludens protein 1 (ZO-1), E-cadherin, occludin) but negatively with the mesenchymal marker N-cadherin and the EMT-inducing proteins Snail and Slug. Subsequently, EMT-dependent increase in tumor migration and invasiveness led to a higher tumor malignancy in NSCLC [49].

A recent study by Han et al. links WWC3 expression to autophagy and apoptosis in NSCLC [50]. WWC3 induces the expression of apoptosis-related molecules while preventing the proliferation of starved or stressed malignant cells. Furthermore, in the presence of WWC3, the activity of caspase 3 and caspase 7 was increased, promoting WWC3 as a regulator of autophagy in starvation-stressed NSCLC [50].

WWC3 was identified to interact with the FERM-domain-containing protein-1 (FRMPD1), which is downregulated in NSCLC [51]. In non-malignant cells, the WWC3–FRMPD1 interaction enhances phosphorylation of LATS1 and Hippo signaling, hindering the transcription of YAP target genes and reducing malignancy. However, in NSCLC, WWC3 and FRMPD1 levels are reduced and Hippo signaling is low, which correlates with advanced tumor–node–metastasis stage, lymph node metastasis and poor prognosis [51].

Furthermore, WWC3 was identified as a pathway connector linking Hippo signaling and the Wnt pathway in NSCLC cells [48]. Earlier studies revealed that the protein Dishevelled 2 (DVL2) binds to and phosphorylates the casein kinase 1ε (CK1ε), negatively regulating Wnt signaling [52]. As WWC3 and CK1ε competitively interact with DVL2, phosphorylation of CK1ε is inhibited by the WWC3–DVL2 interaction and Wnt signaling is enhanced. Simultaneously, LATS1 and Dvl2 interact competitively with WWC3. Thus, the WWC3–DVL2 complex hinders WWC3 activity in Hippo signaling on the one hand, while activating the Wnt pathway on the other hand, thus promoting lung cancer cell proliferation [48].

Wang et al. displayed that overexpression of WWC3 attenuates Wnt signaling in glioma cells in vitro [53]. It is known that the pathway modulator ß-catenin binds to a complex containing T cell factor 4 (TCF4), resulting in enhanced Wnt activity (reviewed in [54]). WWC3 interacts with TCF4, thus preventing the complex formation of TCF4 and ß-catenin and suppressing the Wnt signaling cascade [53]. Hence, WWC3-mediated inhibition of Wnt signaling results in decreased glioma cell proliferation and migration. In contrast, a low WWC3 level forces an increase in Wnt signaling and subsequent proliferation, migration and invasion of glioma cells [53].

Additionally, a study by Yang et al. identified a novel and complex axis regulating WWC3 expression and Hippo signaling in glioma [55]. The authors demonstrated that in glioma cells, the level of the oncogenic Broad-Complex, Tramtrack and Bric a brac (BTB) domain and CNC homolog 2 (BACH2) protein is increased. BACH2 interacts with the fused in sarcoma (FUS) protein, leading to inhibition of a long non-coding RNA (TSLNC8) which would otherwise negatively regulate expression of miR-10b-5p. miR-10b-5p, highly expressed in glioma, inhibits the tumor-suppressive function of WWC3, thus promoting low Hippo pathway activity and enhanced glioma malignancy [55].

A reduced WWC3 expression that correlates with increased cell proliferation and elevated metastasis was also found in GC [56]. Overexpression of WWC3 in GC cell lines led to a reduced synthesis of proteins associated with cell cycle transition (cyclin D1 and cyclin E), enhancing the percentage of cells in G1 phase but reducing percentage of S-phase cells. This indicates that WWC3 blocks the transition of G1 to S phase and thereby inhibits cellular growth in GC. The impact of this mechanism on Hippo signaling is still unknown.

Human schwannomas (SCHW) are associated with mutations or deregulated expression of NF2, a protein known to form a complex with WWC proteins, activating Hippo signaling. Interestingly, studies in Tasmanian devils showed that loss-of-function mutations in WWC3 are linked to a transmissible cancer of Schwann cell origin [57]. This unique cancer type in the Tasmanian devil presents specific genomic alterations which can also be found in human SCHW and neurofibromas, pointing to a putative (NF2-dependent) role of WWC3 in these types of human cancer.

Meng et al. discovered a novel WWC3-related tumor-promoting mechanism that involved the zinc finger E-box binding homeobox 1 (ZEB1), an important driver of tumor growth and metastasis in BC. ZEB1 induces the synthesis of a circular RNA (cirRNA) derived from the *WWC3* gene locus, circWWC3 [58]. An enhanced circWWC3 level, associated with poor prognosis in BC, alters the miRNA expression pattern and subsequently activates the oncogenic Ras signal pathway in BC. Silencing of circWWC3 significantly suppresses the proliferation, migration and invasion of BC cells [58].

## 3. WWC Binding Proteins and Hippo Signaling in Cancer

Beside epigenetic control of WWC gene activity, transcription-related and miRNA-mediated regulation of WWC translation and the dynamic interplay between the WWC proteins and their binding partners are obviously crucial for Hippo signaling in cancer. As WWC proteins share a similar structure with several protein-binding motifs (e.g., WW domains), it is not surprising that in vitro interactions studies have identified a common subset of binding partners, including known Hippo pathway components [15,17]. The interaction of the WWC proteins with LATS1/2, leading to enhanced kinase activity, is especially important for Hippo signaling as this complex formation directly influences YAP nuclear shuttling and transcription of pro-proliferative genes [21]. WWC-mediated LATS1/2 activation was supposed to depend on the simultaneous binding of MST1/2 and subsequent LATS1/2 phosphorylation [21]. However, studies from Moleirinho et al. demonstrated an MST1/2-independent effect of the WWC proteins on LATS1/2 phosphorylation, pointing to an alternative (yet unknown) kinase that is recruited to LATS1/2 via WWC proteins [59].

As other known WWC-binding partners can also interact with LATS1/2 and affect their kinase activity, a highly dynamic regulation of Hippo signaling via WWC-containing dimeric, trimeric or multimeric protein complexes probably occurs in vivo. Extra- and intracellular signals disturbing these balanced protein–protein interactions are likely to affect LATS1/2 activity and Hippo signaling in cancer.

Beside LATS1/2 regulation, Herman et al. described an alternative WWC-dependent control mechanism in Hippo signaling that involves the family of Angiomotin (AMOT) proteins [17]. AMOT proteins are localized to either tight junctions (TJs) or the actin cytoskeleton and are able to recruit YAP and TAZ to these two cytoplasmic compartments independently of their phosphorylation status (reviewed in [60]). In addition, AMOT proteins have a more indirect effect on YAP and TAZ localization as they interact with LATS1/2 kinases and affect their enzymatic activity [61]. The intracellular level of AMOT proteins is controlled via their regulated ubiquitination and subsequent proteasomal degradation [62]. WWC proteins interact with AMOT proteins via their WW domain and internal PPxY motifs, respectively [17]. This interaction protects AMOT proteins from degradation, increasing their intracellular levels. Elevated AMOT concentration, in turn, led to cytosolic YAP accumulation and reduced cell proliferation [17,62]. Therefore, deregulated expression of WWC not only results in low LATS1/2 activity but also in enhanced AMOT degradation, both supporting YAP-mediated cell proliferation. Accordingly, recent studies revealed AMOT proteins as tumor suppressors in glioblastoma, ovarian cancer and lung cancer (reviewed in [63]). However, proliferation and invasion of cancer cells in BC, osteosarcoma, colon cancer, PC, head and neck squamous cell carcinoma, cervical cancer, liver cancer and renal cell cancer correlated with elevated AMOT expression (reviewed in [63]). Thus, the role of AMOT proteins in tumor formation and their interplay with WWC proteins are obviously variable depending on the affected cell type or organ.

Another WWC1-binding partner with an impact on Hippo signaling is the protein tyrosine phosphatase non-receptor type 14 (PTPN14). This cytosolic phosphatase was identified as a possible tumor suppressor, preventing nuclear YAP activity and cell proliferation independently of its enzymatic activity [64]. Later, the interactions between invertebrate *Kibra* and *Pez* (PTPN14 homolog in *Drosophila*) as well as WWC1 and PTPN14 in mammalian cells were described [65]. PTPN14 harbors an internal PPxY motif that mediates the binding to the WW domains of both YAP and WWC1. In vitro studies revealed that WWC1 and PTPN14 can activate LATS1 and Hippo signaling autonomously but also synergistically to suppress cell migration and tumorigenesis (reviewed in [66]).

The WW domains of the WWC proteins obviously fulfill a crucial role in Hippo signaling as they mediate the interaction with several upstream regulators, including LATS1/2, AMOT proteins, DDR1, PTPN14 and Dvl2 (Table 1). Interestingly, all of these proteins are also interaction partners for YAP’s WW domains, indicating a similar binding specificity of the WW domains of YAP and the WWC proteins. These findings model a complex scenario in which a group of PPxY-harboring proteins compete to bind to the WWC and/or YAP WW domains. The imbalance of these temporal- and spatial-regulated protein–protein interactions may have an important impact on Hippo signaling in tumor formation.

Atypical PKC (aPKC), one the first identified binding partners of WWC1 [67], represents a crucial regulator of cell polarity and intracellular transport processes and is frequently overexpressed and activated in many cancer types [68]. All WWC proteins interact with aPKC via a short internal binding motif similar to the aPKC pseudosubstrate region [67,69]. The presence of two aminoterminal WW domains within the WWC proteins mediating LATS1/2 interaction and their common carboxyterminal aPKC-binding motif could theoretically allow the formation of a trimeric LATS1/2–WWC–aPKC complex. However, another model suggests a dynamic and regulated shuttling of the WWC proteins between their LATS1/2-dependent role in Hippo signaling and an aPKC-related impact on cell polarity and cell migration. In cancer, elevated aPKC binding may suppress the activating effect of WWC proteins on Hippo signaling, supporting cell proliferation. In *Drosophila*, the cell polarity regulator *Crumbs* was demonstrated to recruit *Kibra* to apical junctions, repressing Hippo pathway activity [70]. In contrast, Mao et al. could show that in mammals, a CRUMBS3/WWC1 complex enhances Hippo signaling and prevents tumor growth [71]. A similar effect was shown for NF2 which sequesters WWC1 at apical membranes, supporting Hippo pathway activation (reviewed in [72]). The role of a CRUMBS/WWC or a NF2/WWC protein complex in tumor growth and cancer has not been analyzed yet.

Table 1 lists important WWC-binding proteins with a known function in Hippo signaling and different cancer entities.

## 4. Conclusions

Our knowledge on the impact of a deregulated Hippo pathway in cancer is constantly growing and the number of publications dealing with this aspect is increasing from year to year. However, the identification of novel regulatory mechanisms affecting gene transcription, epigenetic gene silencing, miRNA-mediated translation control, dynamic protein–protein interactions and post-translational protein modifications indicate the huge complexity of Hippo signaling under physiological and pathological conditions. Therefore, it is challenging to target key molecules in Hippo signaling for new therapeutic interventions.

The current findings on the link between WWC function and Hippo signaling clearly underline the important role of this protein family in uncontrolled cell proliferation and cancer. Deregulated WWC expression has been found in a variety of human cancers, indicating common effects of low WWC levels on different cell types or tissues. However, the role of WWC proteins as tumor suppressors is still a matter of debate, as some studies demonstrate elevated WWC expression in tumor cells (Table 2). Furthermore, WWC proteins obviously control cancer-related signal transduction at different sites and via multiple protein–protein interactions, increasing the complexity of cell- or tissue-specific WWC function.

Mechanistically, WWC proteins affect Hippo signaling canonically (via LATS1/2 activation) or non-canonically (e.g., via interaction with AMOT and PTPN14) at different steps (Figure 3). In addition, WWC proteins act as pathway connectors linking Hippo and Wnt signaling (Figure 3). Through their interaction with proteins such as aPKC and CRUMBS, WWC proteins may also play a role in the cell polarity-dependent control of Hippo signaling (Figure 3).

It is important to note that most of the experimental approaches including WWC analysis examined the impact of only a single WWC family member in human cancer. Nevertheless, the published data revealed that despite a putative redundant function, genomic deletions (such as WWC1 in BC) or disturbed expression of an individual WWC family member is sufficient to decrease Hippo signaling and force cancer formation. These observations may point to isoform-specific functions of the three WWC proteins in different cells or organs. Alternatively, deregulated expression of a single WWC protein may lead to secondary effects, causing overall imbalanced Hippo signaling. Therefore, future studies should consider expression analysis of the whole WWC family to elucidate synergistic or overlapping effects in different cancer entities. This review focuses on the Hippo-pathway-dependent function of WWC proteins in tumorigenesis. However, WWC proteins might have a YAP/TAZ-independent regulatory role in cell proliferation control and different types of cancer, including leukemia. In line with this, Zhang et al. could demonstrate that WWC1 regulates Aurora kinases, enzymes that are involved in cell division and frequently hyperactivated in leukemia [73].

Human WWC gene mutations that cause deregulated Hippo signaling and tumor growth have not been identified yet. However, the linkage between loss of the WWC1 gene and human breast cancer [28] or the formation of hepatocellular carcinoma in mice harboring an induced hepatic WWC1 and WWC2 knockout [17] gives clear evidence for the in vivo function of the WWC proteins as important Hippo pathway regulators and tumor suppressors. Therefore, WWC gene analysis should be included in future large-scale studies on the cancer genome. In addition, new experimental approaches are necessary to elucidate the dynamic interplay of WWC proteins and their binding proteins to uncover post-translational effects on WWC activity and to determine their role as pathway connectors. More knowledge about the function of WWC proteins will support our understanding of cancer-related processes, allowing identification of novel therapeutic options.

## Figures and Tables

**Figure 1 cancers-13-00306-f001:**
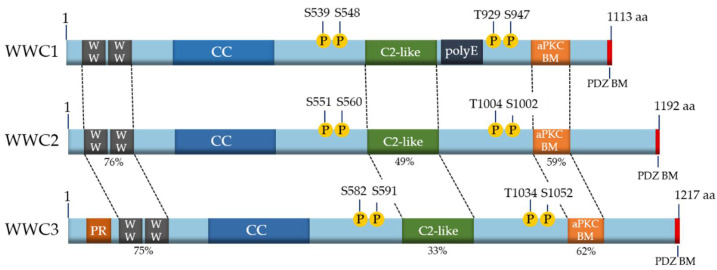
Structure of human WWC proteins. The three members WWC1 (1113 amino acids (aa)), WWC2 (1192 aa) and WWC3 (1217 aa) of the human WWC family share a common structure with two WW domains, a region with potential coiled-coil domains (CC), an internal C2-like domain, an atypical protein kinase C (aPKC)-binding motif (aPKC BM) and a carboxyterminal binding motif for PDZ domain-containing proteins (PDZ BM). A poly glutamic acid stretch (polyE) is unique to WWC1 and a proline-rich region (PR) region is only found in WWC3. WWC proteins can be post-translationally phosphorylated at the indicated conserved sites (P). The sequence similarities of the main protein domains compared to WWC1 are shown as percentages. Modified from [10].

**Figure 2 cancers-13-00306-f002:**
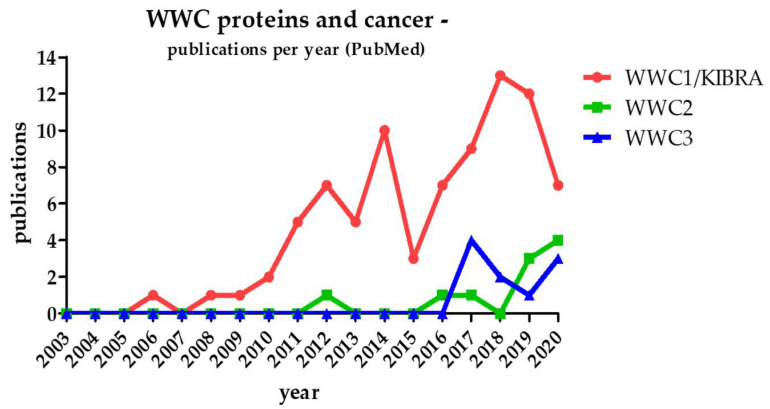
Number of PubMed-listed publications regarding WWC proteins in the context of cancer since the discovery of WWC1 (and KIBRA) in 2003.

**Figure 3 cancers-13-00306-f003:**
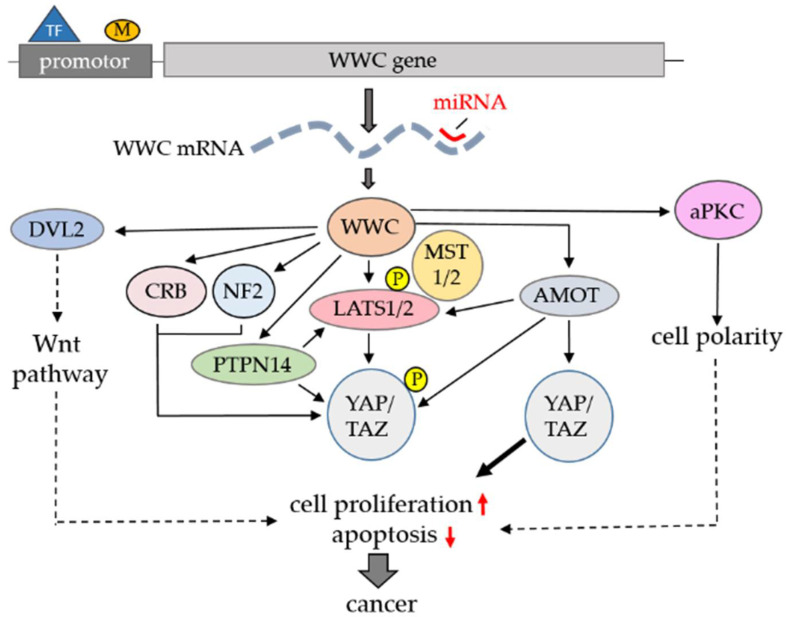
Schematic summary of the regulated expression of WWC proteins and their role in Hippo signaling and cancer. WWC protein activities are controlled at the level of promotor regulation via binding of different transcription factors (TFs) and/or epigenetic DNA methylation (M), miRNA-dependent translation control or specific protein–protein interactions in canonical and non-canonical Hippo signaling. Furthermore, WWC proteins affect alternative pathways, including Wnt signaling or cell polarity-dependent processes, with additional effects on cell proliferation and cancer.

**Table 1 cancers-13-00306-t001:** Selected WWC-binding partners and impact of the protein–protein interaction on Hippo signaling in cancer. BC, breast cancer; GC, gastric cancer; HCC, hepatocellular carcinoma; NSCLC, non-small-cell lung cancer; TNBC, triple-negative breast cancer.

Binding Partner	WW Binding Domain	Effect on Hippo Pathway	Link to Cancer [Reference]
LATS1/2 (WWC1/2/3)	WW domains	Activation	HCC [17]
AMOT (WWC1/2)	WW domains	Activation	HCC [17]
PTPN14 (WWC1)	WW domains	Activation	TNBC [66]
FRMPD1 (WWC3)	PDZ binding motif	Activation	NSCLC [51]
DVL2 (WWC3)	WW domains	Activation	NSCLC [48]
DDR1 (WWC1)	WW domains	Unknown	BC [22]
aPKC (WWC1)	aPKC binding motif	Inhibition	BC [22], GC [31]

**Table 2 cancers-13-00306-t002:** Summarized data about deregulated expression of the different WWC family members in cancer. ALL, acute lymphocytic leukemia; BC, breast cancer; BLC, bladder cancer; ccRCC, clear-cell renal cell carcinoma; CLL, chronic lymphocytic leukemia; CRC, colorectal cancer; GC, gastric cancer; HCC, hepatocellular carcinoma; LUAD, lung adeno carcinoma; NSCLC, non-small-cell lung cancer; OS, osteosarcoma; PAC, pancreas cancer; PC, prostate cancer.

WWC Family Member	Expression Level	Link to Cancer [Reference]
WWC1	Low	BC [22,26,29]
WWC1	High	BC [27]
WWC1	Low	ccRCC [30]
WWC1	High	GC [31,32]
WWC1	High	BLC [33]
WWC1	Low	CRC [34]
WWC1	Low	OS [36]
WWC1	Low	ALL [37]
WWC1	Low	CLL [38]
WWC1	Low	LUAD [40]
WWC1	High	PC [41,42]
WWC2	Low	HCC [17,43]
WWC2	Low	PAC [44]
WWC2	Low	LUAD [45]
WWC2	Low	CRC [46]
WWC3	Low	NSCLC [48,49]
WWC3	Low	Glioma [53,55]
WWC3	Low	GC [56]

## Data Availability

Not applicable.

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
