# Peer review of "WWC Proteins: Important Regulators of Hippo Signaling in Cancer"

_cancers, 2021, doi:10.3390/cancers13020306_

Round 1

Reviewer 1 Report

The authors address the functional significance of WWC proteins in cancer and in the regulation of Hippo signaling. Overall, the review is straightforward. Perspectives and future directions are somewhat lacking. Please provide the full name of WWC and other needed abbreviations. Describe the structural relationship between WWC1, WWC2 and WWC3 and their functional synergism, antagonism and overlaps. Provide a schematic graph to address the aforementioned issues.

Reviewer 2 Report

The review nicely summarizes the relationship between WWC protein family and Hippo-YAP pathway. However, authors should provide more in vivo relevance (not just correlation in patients) between KIBRA and Hippo-YAP pathway to strengthen the impact of the current review.

  1. Provide paragraphs of current evidence that WWC proteins are cancer genes in vivo. In order to be recognized as cancer genes, their manipulation in vivo should be sufficient to control cancer growth, or human mutation should be identified. Moreover, these in vivo cancer growth should mechanistically be linked to Hippo pathway.
  2. If there are no evidence in transgenic or KO mouse indicating WWC proteins are indeed cancer genes linked to Hippo pathway, these limitations should also be described in the manuscript. WWC1 KO mouse has been generated showing defect in neurons. Is there research done in the cancer context?
  3. There are cancer types with no YAP/TAZ. Not only blood cancers but also SCLC, and some NE tumors do not express YAP/TAZ. What are the functions and mechanism of WWC in those YAP/TAZ deficient cancers? Would that dictate the controversial role of WWC in cancer cells? 
  4. In the context of Hippo pathway, KIBRA is well known to be associated with the Crumb, NF2 complex at the apical cell junctions. This point is not well described in the review and illustration as well. Be more precise regarding the known molecular and cellular role of KIBRA in the context of Hippo pathway.

Reviewer 3 Report

This review by Höffken et al., relates to the understudied WWC family and its connection to cancer. The authors summarize the known and predicted roles of the WWC family proteins WWC1, WWC2 and WWC3 in the context of cancer, and their link to the Hippo pathway. Of these the WWC1 encoded protein KIBRA is the most well described for its roles in cancers, and its role as an upstream regulator of the MST1/2 Hippo kinase. The lesser known members of this family are shown to be involved in cancer as well, however, further investigations are expected to describe the details regarding their connections with Hippo pathway.

Major comments:

  1. The ideas presented in this review are very useful to a wide readership. However, the listing of the different cancers in which WWC family proteins are affected makes it difficult to grasp the various ideas presented in the review. For example, the authors present a list of cancers in which Wwc1 function is affected. Wwc1 can act as a tumor suppressor gene (and loss of Wwc1 can  cause YAP activation) or an oncogene in different cancer contexts. This idea needs to be elaborated to show to what extent Hippo pathway is linked to the perturbance, are there examples for both loss and gain of function of Wwc1 leading to YAP activation, if so, what are the specific contexts and the activation mechanisms that are known so far.
  2. Since very little is known about the WWC family protein a comparison of the protein structure would be very informative and useful.

Minor Comments:

1. Did the authors want to add information under the title Introduction? If not, the organization should be 
        1. Introduction

        1.1 WWC proteins and cancer, and so on.

2. Line 196 Typo:

TWSIT1 should be TWIST1

Round 2

Reviewer 2 Report

The authors well-addressed the reviewer's concerns and the manuscript has significantly improved and suitable for publication.